Resource
# Defining the expression of piRNA and transposable elements in *Drosophila* ovarian germline stem cells and somatic support cells

Benjamin Story*, Xing Ma*, Kazue Ishihara*, Hua Li, Kathryn Hall, Allison Peak, Perera Anoja, Jungeun Park, Jeff Haug, Marco Blanchette, Ting Xie

Piwi-interacting RNAs (piRNAs) are important for repressing transposable elements (TEs) and modulating gene expression in germ cells, thereby maintaining genome stability and germ cell function. Although they are also important for maintaining germline stem cells (GSCs) in the *Drosophila* ovary by repressing TEs and preventing DNA damage, piRNA expression has not been investigated in GSCs or their early progeny. Here, we show that the canonical piRNA clusters are more active in GSCs and their early progeny than late germ cells and also identify more than 3,000 new piRNA clusters from deep sequencing data. The increase in piRNAs in GSCs and early progeny can be attributed to both canonical and newly identified piRNA clusters. As expected, piRNA clusters in GSCs, but not those in somatic support cells (SCs), exhibit ping-pong signatures. Surprisingly, GSCs and early progeny express more TE transcripts than late germ cells, suggesting that the increase in piRNA levels may be related to the higher levels of TE transcripts in GSCs and early progeny. GSCs also have higher piRNA levels and lower TE levels than SCs. Furthermore, the 3′ UTRs of 171 mRNA transcripts may produce sense, antisense, or dual-stranded piRNAs. Finally, we show that alternative promoter usage and splicing are frequently used to modulate gene function in GSCs and SCs. Overall, this study has provided important insight into piRNA production and TE repression in GSCs and SCs. The rich information provided by this study will be a beneficial resource to the fields of piRNA biology and germ cell development.

## Introduction

In animals, germ cells are dedicated to faithfully transmitting the genome from generation to generation. The genome contains many heterochromatic regions, which are rich in transposable elements (TEs), including both DNA transposons and retrotransposons. Mobilized TEs can mutate protein-coding genes, regulatory regions, and impair genome stability in germ cells. Piwi-interacting RNAs (piRNAs) are generated to repress TE activity and maintain genome stability, given that disrupting piRNA production causes infertility in *Drosophila* and mice because of DNA damage–induced blockage of germ cell differentiation (Malone & Hannon, 2009; Thomson & Lin, 2009; Khurana & Theurkauf, 2010; Saito & Siomi, 2010; Banisch et al, 2012). The *Drosophila* ovary contains germline stem cells (GSCs) that provide a continuous supply of differentiated germ cells and eventually mature oocytes throughout their lifetimes (Spradling et al, 2011; Xie, 2013). Although piRNA components are required intrinsically for maintaining GSCs in the *Drosophila* ovary (Ma et al, 2014, 2017), piRNAs in GSCs and their immediate progeny have yet to be characterized. This study uses the *Drosophila* ovary as a model to reveal that piRNA expression levels in GSCs and early progeny are higher than in terminally differentiated germ cells and discovers previously unidentified piRNA clusters.

The adult *Drosophila* ovary contains 12–16 ovarioles with each carrying 2–3 GSCs in its germarium at the tip. GSCs continuously generate cystoblasts (CBs) via asymmetric cell division; Bam and Bgcn function as key differentiation regulators driving CBs to form connected 2-cell, 4-cell, 8-cell, and 16-cell cysts via synchronous division as evidenced by mutations in *bam* and *bgcn* completely blocking further CB differentiation and causing accumulation of GSC-/CB–like cells (McKearin & Spradling, 1990; Ohlstein & McKearin, 1997). *bam* is repressed in GSCs by niche-activated BMP signaling but is then expressed in CBs and dividing cysts (Xie & Spradling, 1998; Chen & McKearin, 2003; Song et al, 2004). Constitutive BMP signaling causes the accumulation of GSC-like cells outside the GSC niche (Xie & Spradling, 1998; Chen & McKearin, 2003; Casanueva & Ferguson, 2004; Song et al, 2004). This study investigates piRNA and TE transcriptional profiles using constitutive BMP signaling and *bam/bgcn* mutations to enrich GSCs and CBs, respectively.

---

Stowers Institute for Medical Research, Kansas City, MO, USA

Correspondence: tgx@stowers.org
Present address: Benjamin Story's present address is European Molecular Biology Laboratory, Genome Biology Unit, Heidelberg, Germany
*Benjamin Story, Xing Ma, and Kazue Ishihara contributed equally to this work

Two distinct piRNA pathways exist in the *Drosophila* ovary, the one in the germline uses three PIWI family proteins (Piwi, Aub, and Ago3) and the other in the soma requires only Piwi (Saito et al, 2006; Brennecke et al, 2007; Yin & Lin, 2007; Li et al, 2009; Malone et al, 2009). Soma-derived piRNAs originate from uni-stranded clusters, whereas most germline-derived piRNAs generally originate from dual-stranded clusters. piRNAs are produced by distinct transcriptional mechanisms and processing machineries. In the soma of the *Drosophila* ovary, the piRNA clusters are transcribed from only one DNA strand through the use of specific PolII promoters. These transcripts are then transported to the Yb body and further processed by Piwi and Zucchini into individual uni-stranded piRNAs (Ross et al, 2014; Huang et al, 2017; Yamashiro & Siomi, 2018). In germ cells (presumably nurse cells), large piRNA clusters in heterochromatin regions exhibit an enrichment for H3K9me3, which can recruit the RDC complex and subsequently the Moon–TRF2–TFIIA-S complex to initiate transcription from both strands in a PolII-dependent, but promoter-independent manner (Pane et al, 2011; Zhang et al, 2012, 2014; Mohn et al, 2014; Chen et al, 2016; Andersen et al, 2017). Cuff, a major component of the RDC complex, has been shown to be necessary for the productive expression of piRNA precursors from dual-stranded clusters by interfering with the recruitment of CPSF and preventing transcript degradation by the exonuclease Rat1 (Chen et al, 2016). These transcripts from both DNA strands are then transported to the perinuclear structure known as the nuage, where they are processed to generate primary piRNAs by Piwi and Zucchini (Saito et al, 2006; Brennecke et al, 2007; Qi et al, 2011; Han et al, 2015; Mohn et al, 2015; Hayashi et al, 2016). Aub-bound primary piRNAs then recognize the sense transcripts, and these sense transcripts can be subsequently cleaved by Ago3 at the 10th nucleotide position complementary with the sense piRNAs. Ago3-bound sense piRNAs target complementary antisense transcripts, thus, resulting in further processing that generates antisense piRNAs. This feed-forward piRNA amplification loop is known as the "ping-pong" cycle (Brennecke et al, 2007; Gunawardane et al, 2007). Besides transcripts from piRNA clusters, those from active TEs also serve as templates for the piRNA processing machinery resulting in the generation of mature piRNAs and, thus, simultaneous silencing of TEs.

Consistent with critical roles of piRNAs in selectively silencing TEs and safeguarding genome integrity, mutations in the genes required for piRNA production lead to elevated DNA damage and subsequent checkpoint activation in germ cells, ultimately resulting in female sterility (Chen et al, 2007; Klattenhoff et al, 2007). In addition, piRNA profiling experiments have been performed on whole ovaries, which are enriched in late differentiated germ cells as GSCs and CBs represent a minority of the total population of germ cells, and furthermore, early functional studies of piRNAs have been focused on late germ cells, nurse cells, and oocyte (Brennecke et al, 2007; Klattenhoff et al, 2007; Li et al, 2009; Malone et al, 2009; Zhang et al, 2012). In the *Drosophila* ovary, GSCs continuously generate differentiating CBs, which ultimately give rise to oocytes, and mutations in a GSC will be passed on to all of its future oocytes. In contrast, mutations occurring in late germ cells only affect individual oocytes. Thus, one might expect tighter control over TE expression in GSCs, given the importance of avoiding the continuous transfer of deleterious mutations into future offspring.

Although the piRNA pathway is demonstrated to be critical for GSC maintenance (Ma et al, 2014, 2017), the composition of piRNAs in GSCs is yet to be defined. In this study, we used cultured GSCs, niche cells, and early differentiation-defective *bam*- and *bgcn*-mutant ovaries to define piRNA clusters and composition in GSCs and their early progeny. Interestingly, the levels of TE-targeting piRNAs are significantly higher in GSCs and CBs than those in the whole ovary. To this end, we developed an algorithm that sequentially scans the genome and finds regions with characteristics that identify them as potential piRNA clusters. Finally, we also used RNA sequencing of purified GSCs and enriched CBs in *bam* and *bgcn* mutants to probe TE expression in GSCs and their early progeny. Surprisingly, most of the known soma-specific TEs are also highly expressed in GSCs and early progeny. Therefore, our profiling of piRNAs and transposons in GSCs and their early progeny provides novel insight into the maintenance of genome integrity in GSCs and opens the door for more focused studies in the future.

## Results

### piRNAs are more abundant in GSCs and early GSC progeny than late differentiated germ cells

In the *Drosophila* ovary, two or three GSCs and CBs can be identified by the presence of a spherical spectrosome, and mitotic cysts can be recognized by the presence of a branched fusome (Lin et al, 1994; de Cuevas & Spradling, 1998) (Fig 1A). Both the spectrosome and the fusome are the same germ cell–specific organelles expressing cytoskeletal proteins, including hu li-tai shao (Hts) (Fig 1A). GSCs and CBs can be easily and reliably distinguished from each other: GSCs directly contact cap cells and express pMad but CBs do not (Xie, 2013) (Fig 1A and A'). *nanos-gal4*–driven germ cell–specific expression of the constitutively active BMP type I receptor *thickvein* ($tkv^{M1}$) results in the accumulation of spectrosome-containing GSC-like cells, which are also positive for pMad (Casanueva & Ferguson, 2004) (Fig 1B). Because *bam* and *bgcn* work in a mutually dependent manner to drive the differentiation of CBs into mitotic cysts (McKearin & Spradling, 1990; Ohlstein et al, 2000), *bam*- and *bgcn*-mutant germaria accumulate many more undifferentiated pMad-negative and spectrosome-containing CB-like cells in addition to two or three pMad-positive GSCs (Fig 1C and D). Thus, this study used $tkv^{M1}$-overexpressing ovaries and *bam*-/*bgcn*-mutant ovaries to enrich GSCs and CBs, respectively, for small RNA and TE analysis using deep RNA sequencing.

To compare the levels of piRNAs and TEs in GSCs, CBs, and whole ovaries, total RNA samples were collected from the wild-type, $tkv^{M1}$-expressing, *bam*-mutant, and *bgcn*-mutant ovaries in triplicates. Small RNAs were gel-purified to select for species smaller than 50 nt and libraries were prepared for deep sequencing (see the Materials and Methods section). All small RNA reads were aligned uniquely to the genome (allowing only a single mismatch). For the reads mapped to the *Drosophila* genome, those reads that matched to the sequences of tRNAs, ribosomal RNAs, small nucleolar RNAs, and small nuclear RNAs were excluded from further analysis (Fig S1A). The remaining mapped reads were then classified

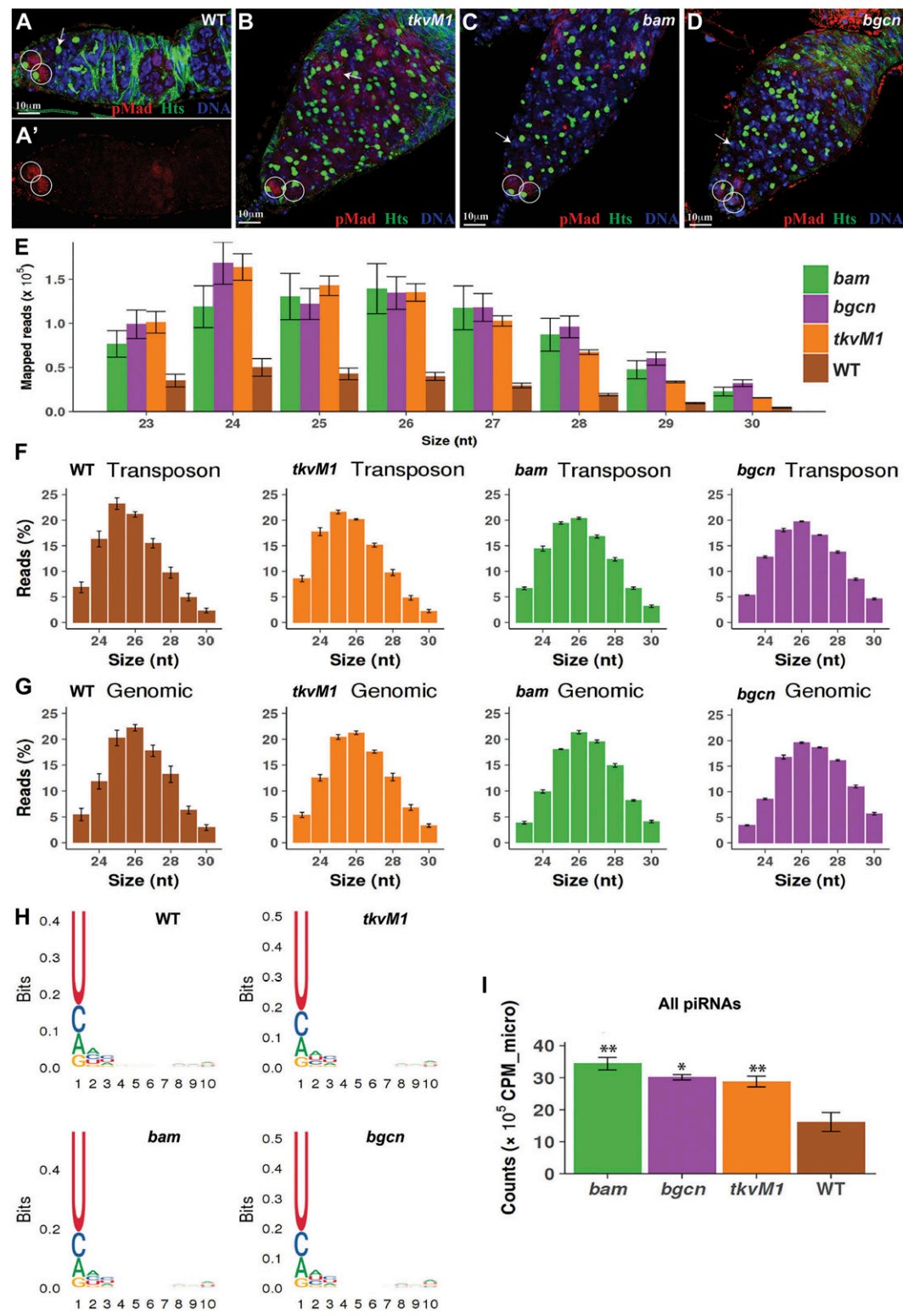

**Figure 1. GSC-/CB-rich ovaries express higher levels of piRNAs than wild-type ovaries.**

Error bars represent SD. **(A, A', B, C, D)** In contrast with the wild-type (A, A') ovary carrying two pMad-positive GSCs (circles), the *tkv*^M1^-expressing (B) ovary accumulates many spectrosome-containing pMad-positive GSC-like cells (arrowhead) besides two pMad-positive endogenous GSCs (circles), whereas *bam*- (C) and *bgcn*-mutant (D) ovaries carry many spectrosome-containing pMad-negative CB-like cells (arrows) besides two pMad-positive endogenous GSCs (circles). **(E)** Number of mapped small RNA-seq reads of the 23–30 nt in length in wild-type, *bam*-mutant, *bgcn*-mutant, and *tkv*^M1^-expressing ovaries. **(F)** Size distribution of piRNA reads mapped to TEs in wild-type and mutant ovaries. **(G)** Size distribution of piRNA reads mapped uniquely to genome in wild-type and mutant ovaries. **(H)** Sequence logo for the first 10 nt of all unique small RNA-seq reads mapped to the genome. The first nucleotide of piRNA reads are biased toward U in wild-type and mutant ovaries. **(I)** GSC-/CB-rich ovaries express more piRNAs than wild-type ovaries.

into three different categories based on their sizes and previous studies (Lau et al, 2009; Wen et al, 2014): siRNAs, miRNAs, and piRNAs (Fig S1A).

In *Drosophila*, RNA hairpin structures trimmed by the Drosha/Pasha complex are loaded onto the Dicer-1/Loqs complex to produce miRNAs if forming imperfect RNA duplexes, or the Dicer-2/R2D2 complex to produce siRNAs if forming perfect RNA duplexes (Ghildiyal & Zamore, 2009). In our analyses, miRNA reads were identified by aligning the 24-nt or shorter reads to the annotated pre-miRNA locations (Ensembl v87 genome annotations), whereas siRNA reads were identified by aligning 22-nt or shorter reads to the annotated endo-siRNA locations or aligning exactly 21-nt reads to transposons (Wen et al, 2014). The remaining 23–30-nt reads that do not align to other small RNA features were considered as piRNAs. Overall miRNA expression levels in wild-type, $tkv^{M1}$-expressing, and *bgcn*-mutant ovaries are comparable, but those in *bam*-mutant ovaries appear to be lower (Fig S1B–F). Those miRNAs also exhibit similar size distribution with 22-nt-long miRNAs being the most abundant species in wild-type, $tkv^{M1}$-expressing, *bam*-, and *bgcn*-mutant ovaries (Fig S1B–F). It is worth noting that 23- and 24-nt-long miRNAs are more abundant in *bgcn*-mutant ovaries than in $tkv^{M1}$-expressing or *bam*-mutant ovaries. In the future, it will be interesting to investigate if Bam and Bgcn are involved in miRNA processing in germ cells (Fig S1E). For siRNAs, $tkv^{M1}$-expressing and *bam*- and *bgcn*-mutant ovaries appear to have less overall siRNAs than wild-type ovaries, and 21-nt-long siRNAs are the most dominant species present in wild-type and mutant ovaries (Fig S1B–E and G). These results suggest that wild-type ovaries have higher overall expression levels of siRNAs than differentiation-defective mutant ovaries, but overall miRNAs are at comparable levels among wild-type and mutant ovaries, excluding *bam* mutants.

The 23–30-nt-long piRNA reads exhibit the known piRNA characteristics (hereafter referred to as piRNAs) with the most abundant lengths of 24–27 nt (Juliano et al, 2011; Banisch et al, 2012; Weick & Miska, 2014; Huang et al, 2017) (Fig 1E). Comprehensive details of the genomic piRNA are provided in the Supplementary Materials (Supplemental Data 1). Although the 23–30-nt-long RNA reads mapped to TEs are more abundant than those mapped to unique genomic locations, the size distribution of piRNAs in wild-type and mutant ovaries is quite similar (Fig 1F and G) (Brennecke et al, 2007). In addition, these 23–30-nt-long RNAs are biased toward a U nucleotide at their first position (Fig 1H). One caveat of this study is that we did not perform a Piwi/Aub/Ago3 pull-down or a detection of 2′-O-methylation at the 3′ end to confirm the authenticity of our identified piRNAs, which we identified based on the length. Three normalization methods, to total small RNA reads, miRNA reads, or siRNA reads, yield consistent results (Figs 1I, S1H and I, and S2G). Interestingly, these $tkv^{M1}$-expressing ovaries containing GSC-like cells have significantly more piRNAs than the wild-type ovaries, suggesting that GSCs have more piRNAs than germ cells at later developmental stages (Figs 1I, S1H, and S2G). In addition, CB-rich *bam*- and *bgcn*-mutant ovaries also express higher piRNA levels than wild-type ovaries, suggesting that early GSC progeny also have more piRNAs than germ cells at later developmental stages (Figs 1I, S1H and I, and S2G). These results, taken together, suggest that GSCs and early

progeny express higher piRNA levels than further differentiated later germ cells.

## Cultured GSCs express significantly more piRNAs than somatic support cells (SCs)

Because these $tkv^{M1}$-overexpressing and *bam-/bgcn*-mutant ovaries also contain various types of somatic cells, including follicle cells, we used our cultured GSCs for the identification of small RNAs. As reported previously (Ma et al, 2017), those GSCs can be expanded in vitro, with ovarian somatic cells, to a large number for molecular analysis, and can also be induced to differentiate into cysts upon Bam expression. Because those cultured GSCs, but not co-cultured somatic cells, express *vas-GFP*, we used FACS to separate GFP-positive GSCs from the co-cultured somatic cells for sequencing mRNAs and small RNAs. Based on normalized mRNA counts (comparable expression levels of *mRps7* and *mRpL47* encoding mitochondrial ribosomal proteins), purified GFP-positive GSCs highly express germ cell–specific genes, *vasa* and *nos*, but the GFP-negative SCs also express low levels of *vasa* and *nos*, ~2–7% of those in germ cells, indicating that there are low levels of GSC contamination in SCs (Fig S2A). In addition, those GSCs also express *put* and *tkv* (encoding type II and type I BMP receptors, respectively, for receiving BMP signal from the somatic niche) and *shg* (encoding E-cadherin for GSC interaction with somatic niche cells) like GSCs in vivo. The in vitro GSC-supporting SCs express the markers for escort cells (such as *tkv* and *fax*), cap cells (such as *shg*, *lamC*, and *dpp*) or both (such as *ptc* and *vkg*), suggesting that they resemble somatic progenitor cells, having the properties of both cap cells and escort cells (Fig 2A). In addition, we also examined the expression of various somatic and germline piRNA pathway components in GSCs and SCs. Known germline-specific piRNA components, *aub*, *ago3*, *vas*, *qin*, *rhi*, and *cuff*, are highly expressed in GSCs, whereas *krimp*, *spn-E*, *tud*, *piwi*, *vret*, *mael*, *armi*, and *Yb* are expressed in both GSCs and SCs (Fig 2B). Surprisingly, somatic piRNA component *Yb* is also expressed in GSCs, which needs further *in vivo* confirmation (Ross et al, 2014). These results indicate that piRNA pathway components are properly expressed in cultured GSCs and SCs.

Similar to small RNAs in the ovary, more than 95% of small RNA reads from the cultured GSCs and SCs belong to piRNAs, miRNAs, and siRNAs (Fig S2B). In this set of experiments, synthetic spike-in RNA standards were added to normalize the small RNA library sizes. miRNAs in GSCs and SCs are overall shifted to smaller sizes compared than those in the ovaries of wild-type and mutants, although 22-nt-long miRNAs are still the dominant size (Fig S2C and D). In addition, GSCs have less miRNAs and siRNAs than SCs (Fig S2E and F). piRNA size distributions in GSCs and SCs are similar to those in the wild-type and mutant ovaries (Fig 2C–E). Both TE-derived and unique genome-derived piRNAs are more abundant in GSCs than in SCs, further supporting the notion that piRNAs have more important roles in germ cells than in somatic cells (Fig 2D and E). Those purified GSCs have more abundant piRNAs than the wild-type and mutant ovaries (Fig S2F). Consistently, the piRNAs in these cultured GSCs have the bias toward U at the first nucleotide. Taken together, our results suggest that GSCs and early progeny have more abundant piRNAs, and thus, might also exert tighter piRNA-mediated repression of TEs than late germ cells and SCs.

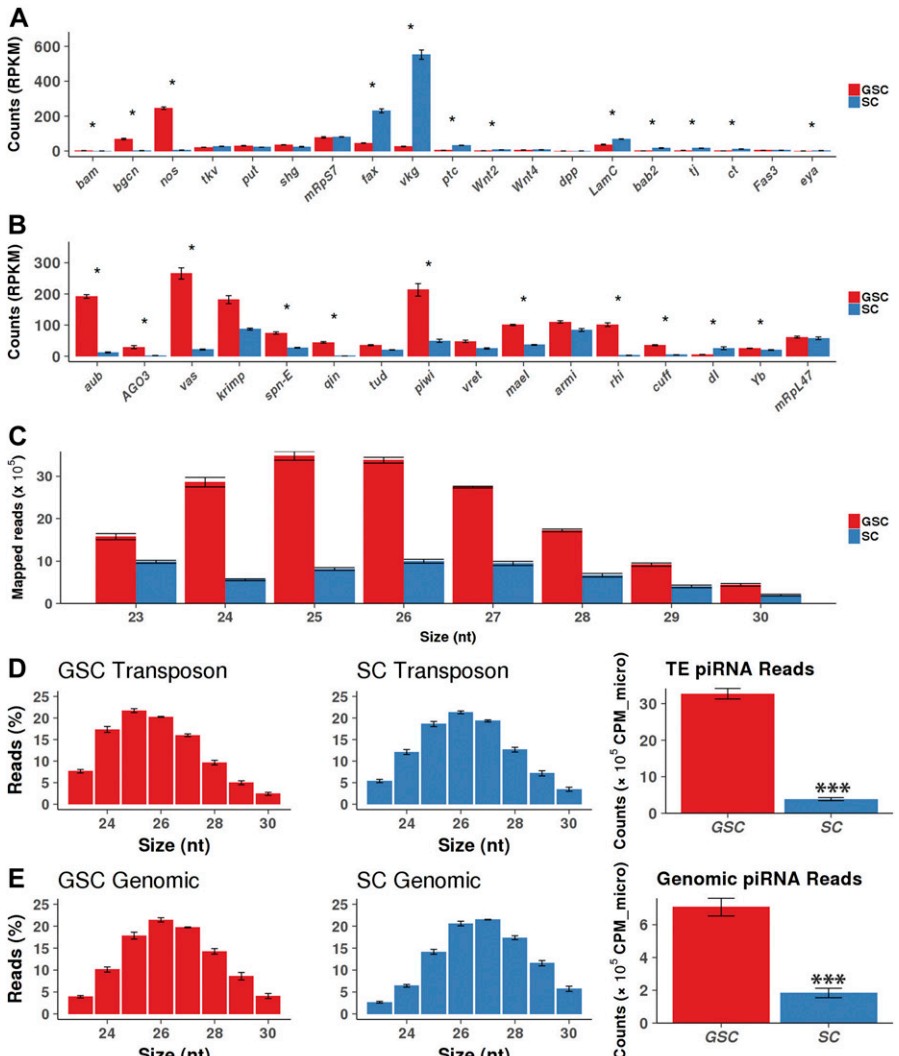

**Figure 2. Cultured GSCs express more piRNAs than their support niche cells.**
Error bars represent SD. **(A)** mRNA-seq results show that in vitro cultured GSCs express known GSC markers, whereas supporting somatic cells (SCs) express known markers for cap cells and escort cells. **(B)** mRNA-seq results show that cultured GSCs and SCs express various piRNA components. **(C)** The size distribution of mapped small RNA reads from the purified cultured GSCs and SCs. **(D)** Size distribution of piRNA reads mapped to TEs in GSCs and SCs. GSCs express more TE-derived piRNAs than SCs. **(E)** Size distribution of piRNA reads mapped uniquely to genome in GSCs and SCs. GSCs express significantly more unique genomic piRNAs than SCs.

## Canonical piRNA clusters show distinct expression patterns in GSCs and SCs

Given the observed higher piRNA expression levels in GSC-/CB-enriched ovaries and cultured GSCs, we decided to investigate how changes in overall piRNA expression levels could be related to changes in individual piRNA clusters. To do this, we examined the expression levels of canonical piRNA clusters as previously defined (Brennecke et al, 2007, 2008). All the known major piRNA clusters show lower expression in the SCs than the wild-type and GSC/CB tumorous ovaries except cluster 8/*flamenco* (Fig 3). Consistent with the idea that the *flamenco* cluster is primarily transcribed in the ovarian soma, its expression is higher in the SCs than GSCs, but surprisingly, it is also expressed at moderate levels in purified GSCs, suggesting that it is not truly soma-specific (Fig 3A). Most of other canonical piRNA clusters, including 1/42AB, 2/20A, 5/38C, 6/80E-F, 9/20B, and 11/100F, are primarily expressed in GSCs (Fig 3B–J). Taken together, these results suggest that transcripts from known piRNA clusters are differentially expressed in GSCs and SCs.

In germ cells, piRNAs can be amplified, and TEs can be effectively silenced through the "ping-pong" cycle (Brennecke et al, 2007; Gunawardane et al, 2007). We used a recently developed software package to scan for potential piRNA-derived ping-pong signals across the genome (Uhrig & Klein, 2018). For all the canonical piRNA clusters, the overall number of ping-pong signatures is high in GSCs, and is extremely low in SCs, potentially due to low-level GSC contamination, indicating that the ping-pong amplification cycle is only active in GSCs (Fig 3K). Among them, cluster5/38C and cluster1/42AB show the highest ping-pong signatures in GSCs, whereas the cluster2/20A exhibits the lowest ping-pong signature in GSCs (Fig 3K'). These results further support the notion that the ping-pong amplification pathway is active in GSCs but not in SCs.

## Identification of new piRNA clusters in GSCs and SCs

Although known piRNA clusters contribute to a global increase of piRNAs in GSCs and CBs, we wanted to investigate if other previously un-annotated clusters also contribute to this up-regulation. To this end, we developed a sliding window algorithm that scanned the

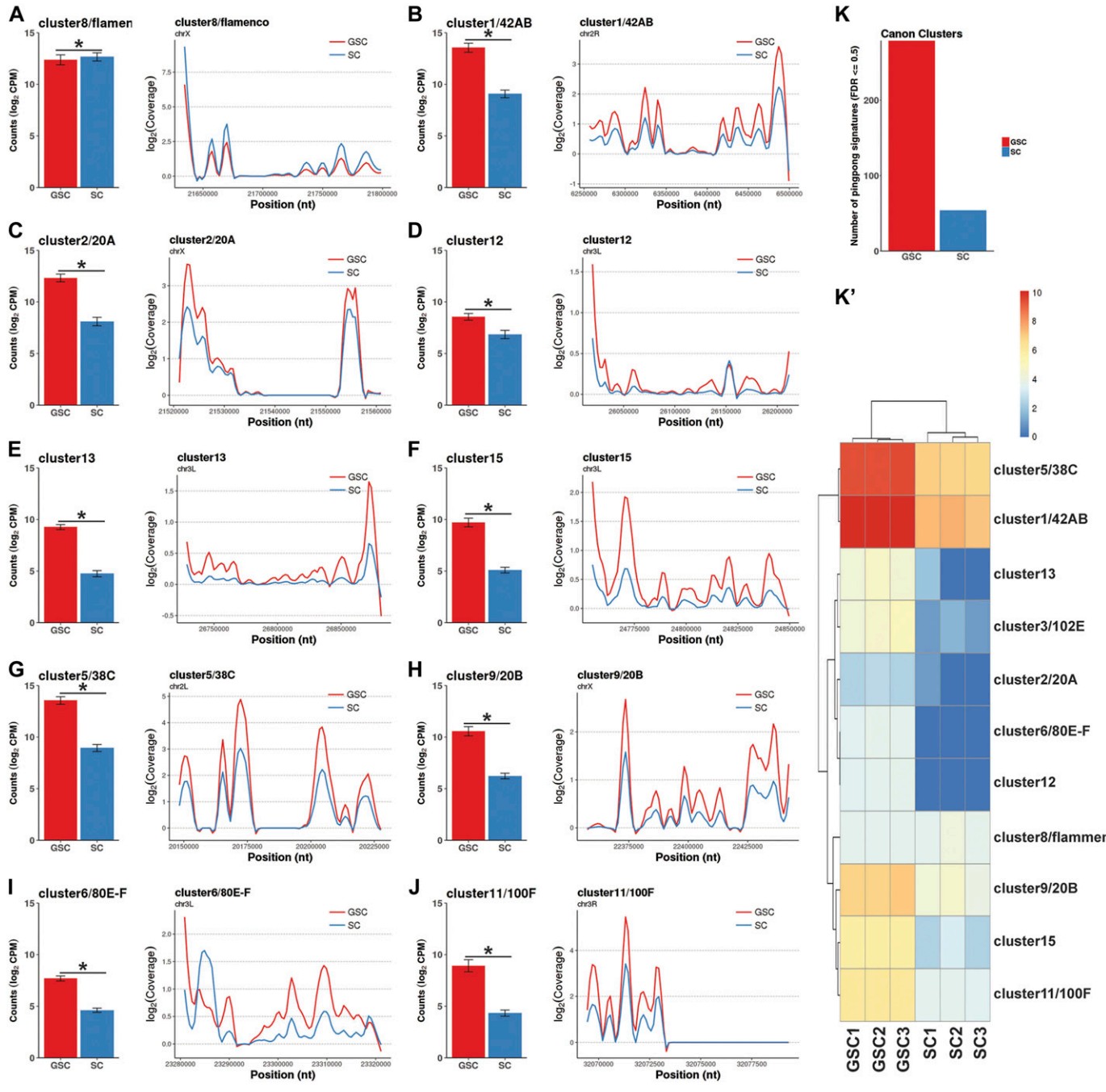

**Figure 3. Most of canonical piRNA clusters show significantly higher expression levels in cultured GSCs than in SCs.**
Each panel contains a bar plot of the normalized read counts (left) and a loess-smoothed (span = 0.1, size = 1) coverage plot (right). Error bars represent 95% confidence intervals. **(A)** Cluster 8 is expressed less in GSCs than in SCs. **(B, C, D, E, F, G, H, I, J)** Clusters 1/42AB, 2/20A, 5/38C, 6/80E-F, 9/20B, 11/100F, 12, 13, and 15 are expressed significantly less in SCs than in GSCs. **(K, K′)** Numbers of ping-pong signatures for all the canonical clusters (K) and heat map for ping-pong signatures of individual clusters (K′) in GSCs and SCs.

genome for regions that were at least 2,000 nt in size, more than 1,000 nt away from neighboring clusters, and that exhibited high densities of unique piRNAs. Our method successfully identified piRNA clusters that overlapped with more than 97% of the previously identified clusters, validating the effectiveness of our new algorithm (Brennecke et al, 2007; Malone et al, 2009). We also provide herein detailed information about the genomic coordinates of

all piRNA clusters identified and their overlaps with previous datasets (Table S1). Interestingly, these newly identified piRNA clusters are distributed throughout the genome but show some bias toward heterochromatic regions such as telomeres and centromeres (Fig S3). Our algorithm has successfully identified 3,365 potential new piRNA clusters, 2,829 of which show relatively high expression levels in wild-type ovaries, mutant tumorous ovaries, GSCs, and SCs (Fig 4A). Among

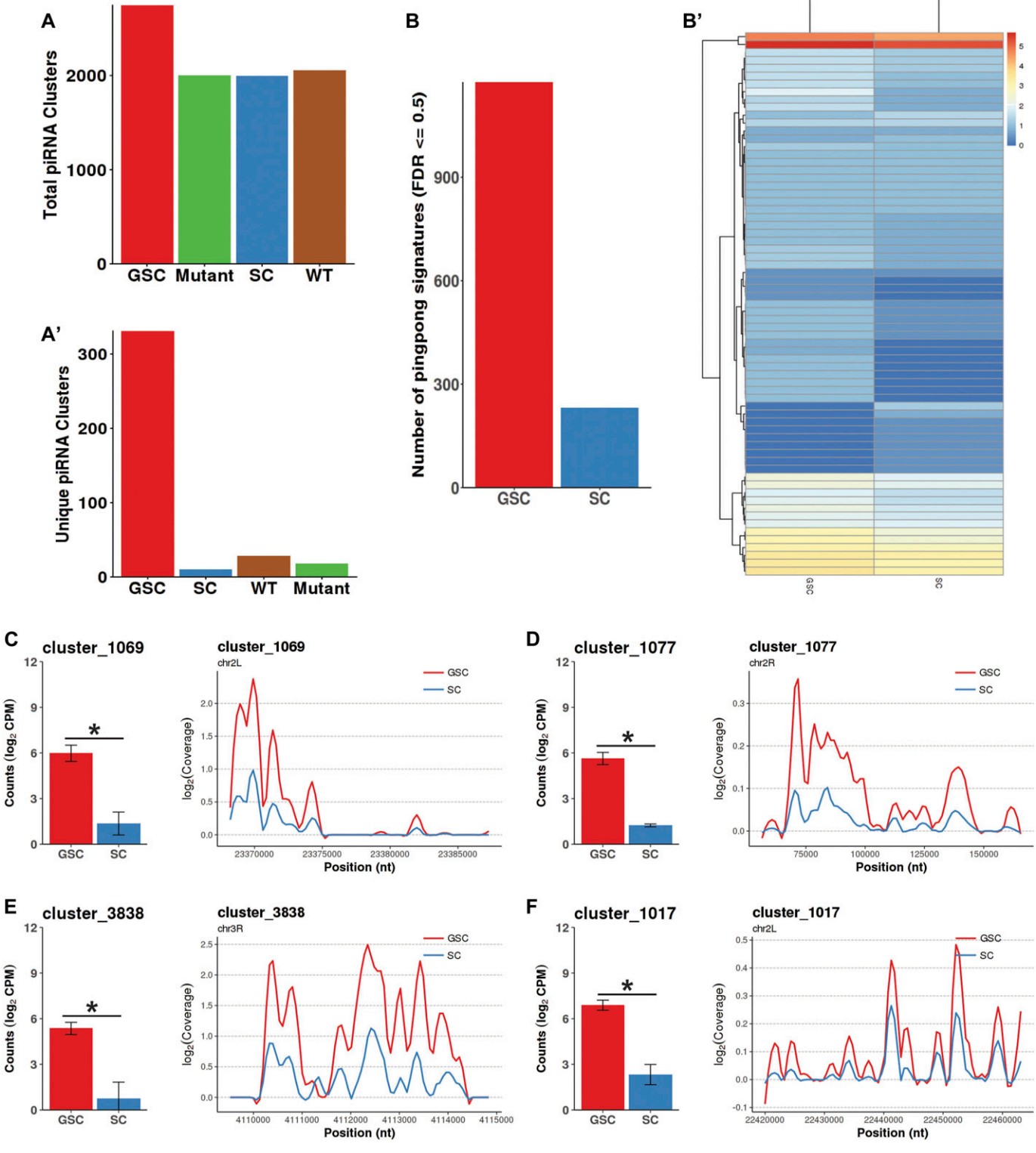

**Figure 4. Newly identified piRNA clusters also show different expression levels in GSCs, SCs, GSC-/CB-rich, and wild-type ovaries.**
**(A, A′)** total (A) and uniquely (A′) expressed (average CPM > 20) new piRNA clusters identified in wild-type, tumorous ovaries, GSCs, and SCs. **(B, B′)** The numbers of ping-pong signatures found consistently in across all replicates for new clusters (B) and a heat map of the counts of ping-pong signatures found in new clusters per sample (B′) for GSCs and SCs (FDR ≤ 0.5). **(C, D, E, F)** Each panel contains a bar plot of the normalized read counts (left) and a loess-smoothed (span = 0.1, size = 1) coverage plot (right). Error bars represent 95% confidence intervals. Clusters 1,069, 1,077, 3,838, and 1,017 show lower expression in SCs than in GSCs.

the newly identified clusters, 18, 28, 10, and 331 new piRNA clusters are uniquely expressed in wild-type ovaries, mutant tumorous ovaries, GSCs, and SCs, respectively (Fig 4A'). Of the newly identified piRNA clusters, 81 are up-regulated, whereas 58 are down-regulated, in the ovaries carrying only GSC-/CB-like cells compared with the wild-type ovaries (false discovery rate [FDR] ≤ 0.05 and fold-change ≥ 2) (Fig 4). This proportion of up-regulated clusters versus down-regulated clusters is similar to the ratio encountered for the canonical clusters. Only 429 of the 3,356 new clusters are located in the regions that do not overlap with any annotated genes. Of these clusters, 195 are significantly up-regulated in GSCs relative to SCs, indicating that most of the newly identified piRNA clusters exhibit higher expression in GSCs relative to SCs (FDR ≤ 0.05 and fold-change ≥ 2). Similar to the canonical piRNA clusters, the overall number of ping-pong signatures for our newly defined piRNA clusters is much higher in GSCs than in SCs (Fig 4B). Among them, most of the new clusters with ping-pong signatures are primarily expressed in GSCs (Fig 4B'). For example, clusters 1,069, 1,077, 3,838, and 1,017 are expressed in GSCs at higher levels than in SCs (Fig 4C–F). Therefore, this study has identified more than 3,000 new piRNA clusters and shown that new piRNA clusters exhibiting more ping-pong signatures tend to have higher expression in GSCs relative to SCs.

## TE transcripts appear to be more abundantly expressed in GSCs/CBs than late germ cells

Because the major function of piRNAs is to repress TEs in the cell, we then investigated if GSCs/CBs have less TE transcripts than late stages of germ cells by sequencing mRNAs from GSC/CB tumorous ovaries, wild-type ovaries, cultured GSCs, and SCs. Surprisingly, the GSC/CB tumorous ovaries exhibit significantly higher levels of all TE transcripts compared with wild-type ovaries (Fig 5A). Among the mutant ovaries, the *bgcn*-mutant ovaries show less up-regulation of TE transcripts than the *bam*-mutant and *tkv*^M1^-overexpressing ones (Fig 5A). We then further investigated if higher TE transcript levels in GSCs/CBs can be attributed to particular TEs by examining the expression of individual TE elements in the wild-type and mutant ovaries. The results from the heat map show that all the known germline-enriched TEs are dramatically up-regulated in the mutant ovaries compared with those in the control, and similarly sense piRNAs produced by these TEs are also up-regulated, suggesting that higher piRNA expression levels in GSCs and CBs are due to high TE expression levels (Fig 5B and B'). In addition, the TEs known to be expressed in both germline and soma, as well as their corresponding piRNA levels, are also drastically up-regulated in the mutant ovaries (Fig 5B and B'). Surprisingly, the previously defined soma-specific TEs are also up-regulated in the mutant ovaries, as

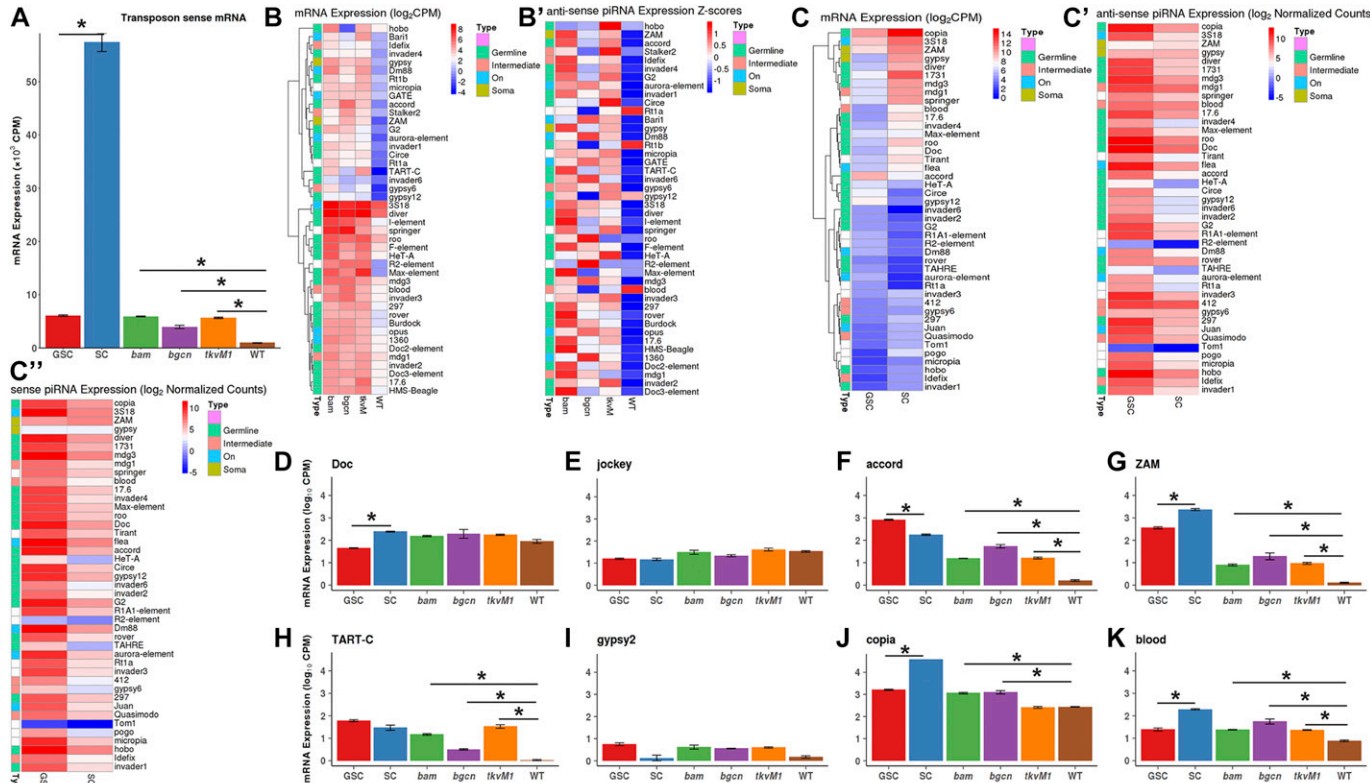

**Figure 5. TE expression levels in GSCs and SCs compared with mutant and wild-type ovaries.**
Error bars represent the standard error of the mean. **(A)** Relative TE expression levels in GSCs, SCs, GSC-/CB-rich, and wild-type ovaries. **(B, B')** Comparison of TE transcript (B) and antisense piRNA (B') expression levels among GSC-rich, CB-rich, and wild-type ovaries. **(C, C', C'')** Comparison of TE transcript (C), antisense piRNA (C'), and sense piRNA (C'') expression levels between GSCs and SCs. **(D, E, F, G, H, I, J, K)** Comparison of the expression levels of TEs, *Doc* (D), *jockey* (E), *accord* (F), *ZAM* (G), *TART-C* (H), *gypsy-2* (I), *copia* (J), and *blood* (K) in GSCs, SCs, GSC-/CB-rich and wild-type ovaries.

are their corresponding piRNA expression levels (Fig 5B and B'). These results indicate that higher piRNA expression levels in GSCs and CBs are likely a consequence of higher TE expression levels, but not related to an increase in TE repression.

Then, we examined the mRNA expression levels of TEs and piRNAs in cultured GSCs and SCs. Overall, cultured GSCs also express significantly higher TE transcripts than the wild-type ovaries, whereas SCs express even higher levels of TEs than GSCs (Fig 5A). The elevated levels of TE transcript in SCs can mostly be attributed to nine elements, including *ZAM*, *gypsy*, *copia*, *diver*, *1731*, *mdg3*, and *mdg1* (Fig 5C). In contrast, *copia*, *3S18*, *ZAM*, *accord*, and *gypsy12* are the most abundantly expressed TEs in cultured GSCs (Fig 5C). Finally, piRNA expression levels of both sense and antisense piRNAs for individual TEs are not correlated with their corresponding mRNA expression in either GSCs or SCs (Fig 5C and C"). Interestingly, the known soma-specific TEs, *gypsy* and *ZAM,* are also expressed in GSCs in addition to being highly expressed in SCs, whereas the known germline-specific TEs, including *accord*, *Doc*, *TART-C*, and *copia*, are also expressed in SCs in addition to germ cells (Fig 5D–K). It is likely that *Doc*, *jockey*, and *copia* are expressed at comparable levels in germ cells of different developmental stages, whereas *TART-C*, *gypsy-2*, *accord*, *ZAM*, and *blood* are expressed at higher levels in GSCs and early progeny than germ cells of advanced differentiated stages (Fig 5D–K). Along with earlier observations, these results show that GSCs/CBs have higher TE transcripts and piRNA levels than late stage germ cells.

## piRNAs derived from 3' UTRs might be involved in the regulation of SC- and GSC-specific gene expression

It has been previously reported that piRNAs can be produced from 3' UTRs of protein-coding mRNAs, such as *traffic jam* (*tj*) (Robine et al, 2009; Saito et al, 2009). Interestingly, 454 gene transcripts can produce piRNAs from their 3' UTRs in GSCs and SCs: 261, 169, and 24 gene transcripts produce sense, antisense, and both sense and antisense piRNAs, respectively (Fig 6A–F'). For example, *CG15628* and *ari-1* express 3' UTR-derived antisense piRNAs in GSCs and SCs (Fig 6A and B), whereas *CG3812* express 3' UTR-derived sense piRNAs in GSCs (Fig 6C). Among the 261 gene transcripts producing sense piRNAs, 75 of them show significantly higher expression levels in GSCs than in SCs (Fig 6C and Table S2). Among the 75 gene transcripts showing differential sense piRNA expression between GSCs and SCs, 39 of them have transposon fragments in their 3' UTRs, and the other 36 do not (Fig 6G). Interestingly, for the gene transcripts producing antisense piRNAs, 109 of them exhibit differential piRNA expression between GSCs and SCs: 108 of them show higher piRNA expression levels in GSCs, whereas only a single gene exhibits higher piRNA expression levels in SCs (Fig 6A and B and Table S3). Among the 109 gene transcripts showing differential antisense piRNA expression between GSCs and SCs, 63 of them have transposon fragments in their 3' UTRs, and the remaining 46 do not (Fig 6G). For the gene transcripts producing both sense and antisense piRNAs, 12 of them show significantly higher piRNA expression levels in GSCs than in SCs (Fig 6D–F'). These results show that piRNA production from 3' UTRs is regulated in a cell type–dependent manner.

Next, we examined if there was any correlation between piRNA and mRNA levels in GSCs and SCs. For the gene transcripts producing

only antisense piRNAs, 32 of them show differential mRNA expression between GSCs and SCs (Fig 6H and Table S4). Interestingly, 9 and 23 genes show the expression changes in the same direction or the opposite direction for piRNAs and mRNAs, respectively (Tables S3 and S4). For the gene transcripts producing only sense piRNAs, 18 of them exhibit differential expression between GSCs and SCs: 10 and 8 show the expression changes in the same direction or the opposite direction for piRNAs and mRNAs, respectively (Fig 6H and Tables S2 and S4). These results suggest that the differential piRNA production via 3' UTRs might regulate germline- or soma-specific gene expression via complex mechanisms.

## Alternative promoter usage and mRNA splicing are two mechanisms for modulating the production of different protein isoforms in GSCs and SCs

Although microarray-based gene profiling for purified GSCs has been performed (Kai et al, 2005), RNA-seq has not been previously performed on purified GSCs. Our RNA sequencing results have further reaffirmed the results of previous gene expression studies (based on gene ontology [GO] term enrichment analysis for the genes expressed in GSCs). Here, we compared gene expression differences between the cultured GSCs and SCs. Interestingly, SC-enriched genes are often involved in the regulation of cell–cell adhesion, extracellular matrix and cell–cell communications, further underscoring the importance of somatic cells supporting the development of GSCs and progeny (Fig S4A). In contrast, GSC-enriched genes include the genes important for signal reception, RNA regulation, and chromosomal organization, further supporting the notion that the development of GSCs and their progeny is dependent on both signals from SCs as well as transcriptional and posttranscriptional regulation (Fig S4B). The enrichment of the P granules, pole plasm, and ribonucleoprotein GO terms supports this conclusion as they are related to the regulation of RNA stability and translation.

Among the genes expressed in both GSCs and SCs, 102 genes use alternative promoters to produce the different transcripts with distinct 5' ends in their mRNAs in GSCs and SCs, which can result in different 5' UTRs and/or protein N termini (Table S5). For example, *tkv* and *nuf* use different promoters in GSCs and SCs to generate mRNAs with distinct 5' ends, producing proteins with longer N termini in GSCs than in SCs (Fig 7A and B). Based on pMad and *Dad-lacZ* reporter, BMP signaling is only active in GSCs and newly formed CBs, but not in somatic niche cells, which also express all BMP receptors and downstream components. Excitingly, our results here show that GSCs and SCs express different *tkv* isoforms: the Tkv expressed by GSCs contains a signal peptide, whereas the Tkv expressed in SCs lacks most of it. The signal peptide is known to be important for proper transmembrane receptor presentation on the cytoplasmic membrane. The different Tkv isoforms in GSCs and SCs could offer an explanation as to why only GSCs, but not SCs, are responsive to the niche-derived BMP signal. Our results suggest that alternative promoter usage in GSCs and SCs could function as a mechanism for regulating gene expression.

Alternative splicing can also result in the generation of multiple isoforms from a single gene locus with different coding sequences

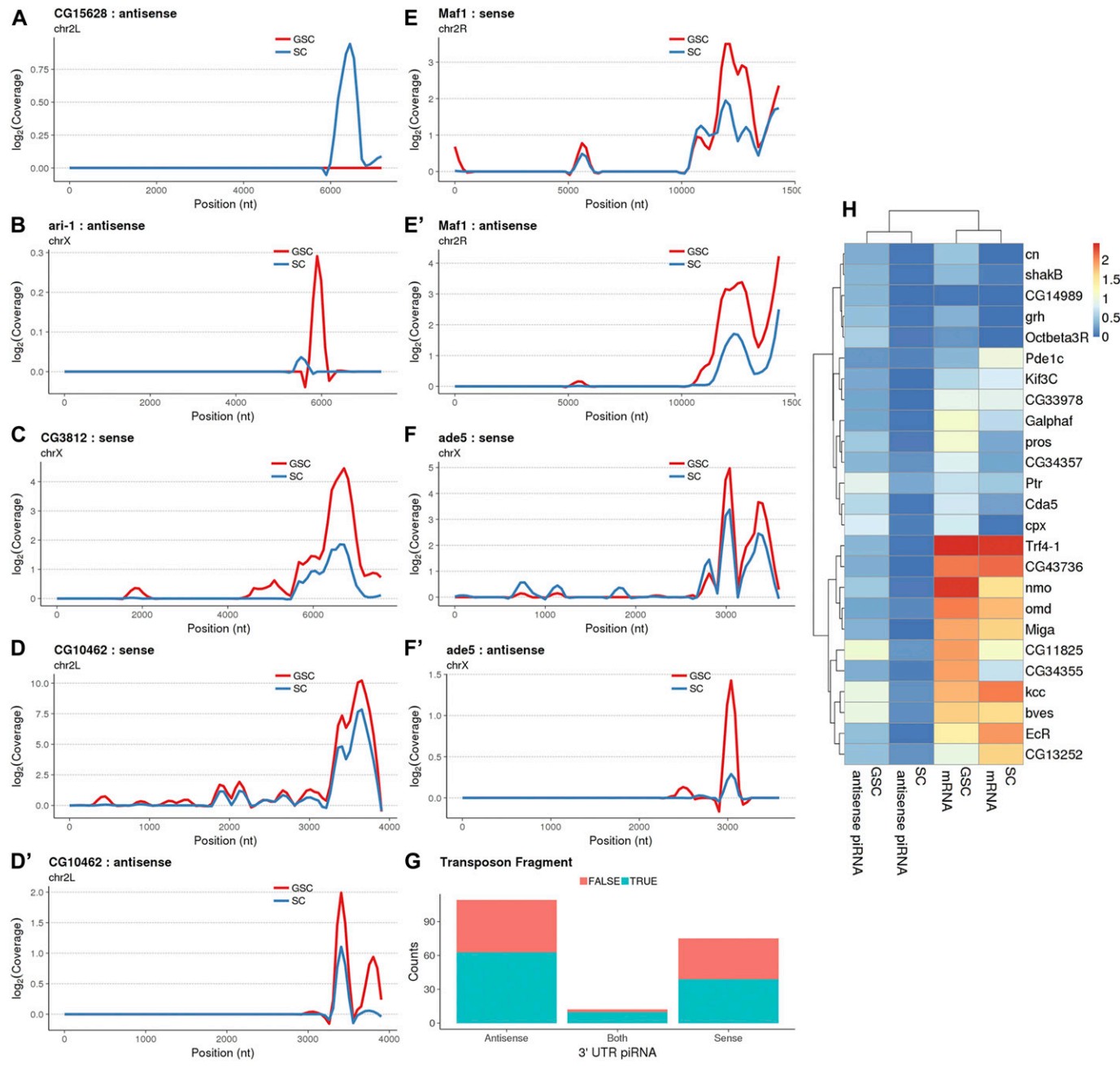

**Figure 6. 3′ UTR-derived sense and antisense piRNAs in GSCs and SCs.**
**(A, B)** Loess-smoothed (span = 0.1, size = 2) coverage plots for 3′ UTR-derived antisense piRNAs from *CG15628* (A) and *ari-1* (B). **(C)** Coverage plot for 3′ UTR-derived sense piRNAs from *CG3812*. **(D, D′, E, E′, F, F′)** Both 3′ UTR-derived sense and antisense piRNAs from *CG10462, Maf1, and ade5* are higher in GSCs than in SCs. **(G)** Distribution of transposon fragments located in the 3′ UTRs of genes exhibiting significant 3′ UTR-derived antisense, from both strands, or sense piRNAs. **(H)** Comparison of expression levels of 3′ UTR-derived antisense piRNAs and their corresponding mRNAs in GSCs and SCs.

or mRNAs with different 3′ UTRs (allowing for the control of gene expression). Our results show that approximately 634 genes exhibit different splicing patterns in GSCs and SCs (Table S5). Interestingly, GSCs sometimes produce mRNAs encoding longer protein isoforms than SCs. For example, *hts* and *pop2* generate proteins with longer C termini in GSCs than in SCs via alternative splicing (Fig 7C and D). The SC-expressed 718-aa-long Hts is known to be localized on the follicle cell membrane, whereas the GSC-expressed Hts with a long

C terminus has been shown to be cleaved into different parts and is localized to fusomes and ring canals in germ cells (Whittaker et al, 1999) (Fig 7C). Pop2 has recently been shown to be important for GSC maintenance and lineage differentiation by functioning as a component in the deadenylase CCR4–NOT complex (Fu et al, 2015; Newton et al, 2015). The GSC-specific Pop2 isoform has a longer C terminus than SC-expressing one (Fig 7D). For *RpL11*, the SC-expressing mRNA has a longer 3′ UTR than the GSC-expressing one, but they include the

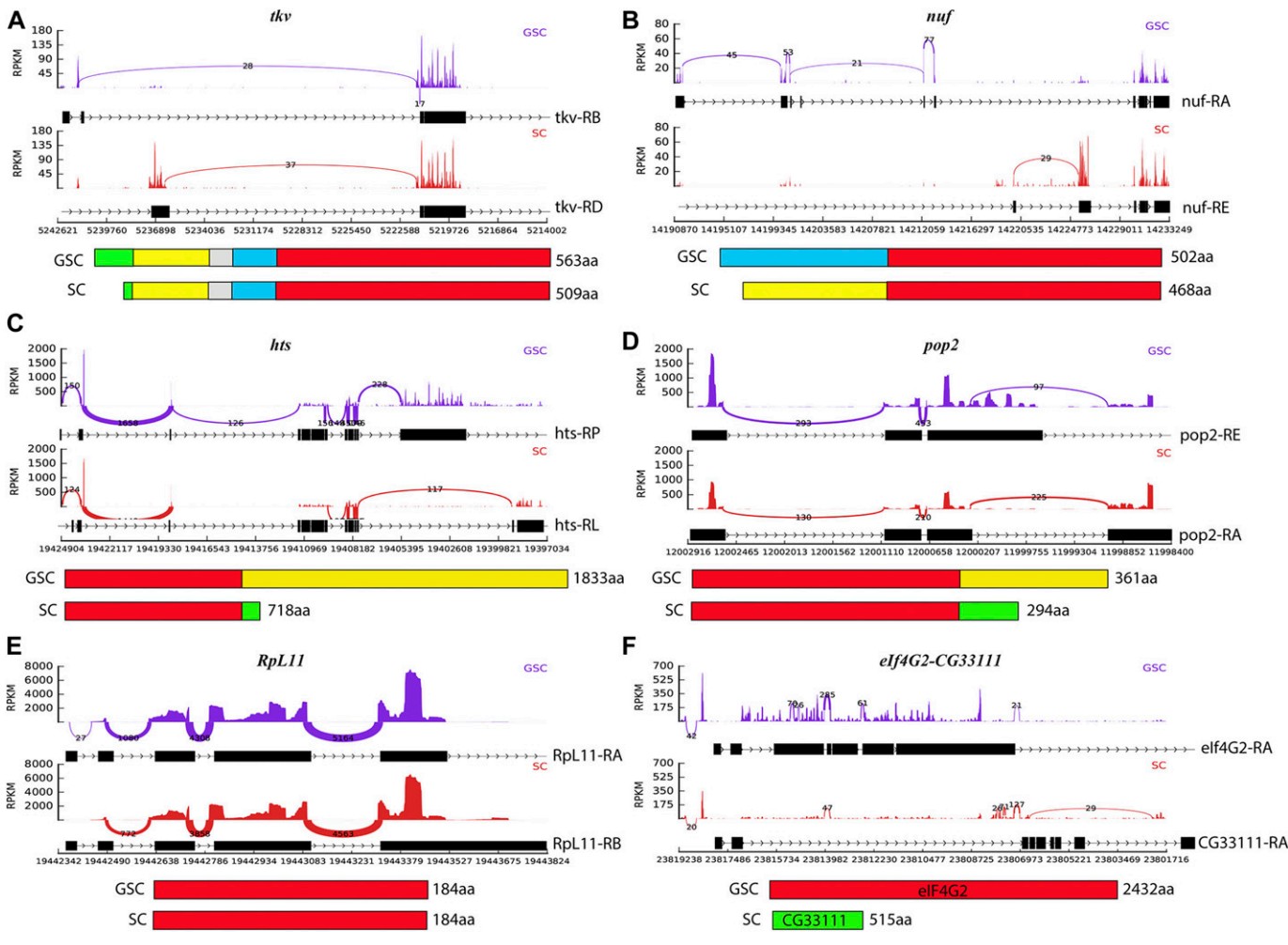

**Figure 7. Differential promoter usage and alternative splicing between GSCs and SCs.**
In each panel, the top Sashimi plots visualize coverage and junction spanning reads derived from the mRNA sequencing of GSCs and SCs, whereas the bottom parts are diagrams of protein structures. **(A, B)** Alternative promoter usage for *tkv* (A) and *nuf* (B) genes produces proteins with distinct N termini in GSCs and SCs. **(C, D)** Alternative splicing for *hts* (C) and *pop2* (D) at the 3′ end generates proteins with distinct C-termini in GSCs and SCs. **(E)** Alternative splicing for *RpL11* results in mRNAs with the same protein-coding capacity but with different 3′ UTRs in GSCs and SCs. **(F)** Alternative splicing results in mRNAs coding for completely different proteins, eIF4G2 and CG33111, in GSCs and SCs.

same coding sequence (Fig 7E). *eIF4G2* and *CG33111* share the first two exons but encode completely different proteins. Because of alternative splicing, only GSCs express the eIF4G2 protein, whereas SCs express the CG33111 protein (Fig 7F). eIF4G2 has been shown to be important for spermatogenesis via translational regulation (Baker & Fuller, 2007; Ghosh & Lasko, 2015). Therefore, alternative splicing functions as a mechanism to modulate both gene function and regulation in the *Drosophila* ovary, especially in regards to GSCs and niche cells.

## Discussion

piRNAs have recently been subjected to extensive studies in the *Drosophila* ovary for their roles in repressing TEs in germ cells and somatic cells (Khurana & Theurkauf, 2010; Juliano et al, 2011; Banisch et al, 2012; Handler et al, 2013; Weick & Miska, 2014; Huang

et al, 2017; Yamashiro & Siomi, 2018). Although piRNA pathway components have been shown to be critical for GSC maintenance (Ma et al, 2014, 2017), piRNA expression in GSCs has never been compared with other stages of germ or niche cells. In this study, our results have shown that GSCs and their early progeny express higher levels of piRNAs while simultaneously exhibiting higher levels of TE transcripts than germ cells of later stages. In addition, we developed a new algorithm to identify piRNA clusters. Furthermore, we revealed that GSCs and SCs express different levels of 3′ UTR-derived piRNAs. Finally, we have also shown that GSCs and SCs use alternative promoters and splicing modulation to control differential gene expression (DGE). Therefore, our results suggest that GSCs and their early progeny regulate piRNA production and/or function differently from late differentiated germ cells as well as somatic niche cells. The datasets generated by this study are a valuable resource for further investigation of piRNA function, TE repression, and DGE in GSCs, early progeny, and somatic niche cells.

## GSCs and their progeny express both higher levels of piRNAs and TE transcripts than late germ cells in the Drosophila ovary

In this study, we sequenced small RNAs from wild-type ovaries, GSC-/CB-rich ovaries and cultured GSCs and niche cells. As expected, most of the abundant piRNAs come from the previously defined canonical clusters. Interestingly, most canonical clusters are up-regulated in GSCs and early progeny in comparison with late germ cells in the *Drosophila* ovary, including *flamenco*. Although *flamenco* has been previously defined as a somatic cell–specific cluster, our results also show that it is also expressed at moderate levels in GSCs and perhaps even in early progeny. Our new algorithm has successfully identified the canonical piRNA clusters and 3,356 new piRNA clusters. Similarly, most of the newly identified piRNA clusters exhibit significantly up-regulated expression in GSCs and early progeny compared with late germ cells. In addition, although most of them are expressed in cultured niche cells and GSCs, some of them are only restricted to germ cells or somatic cells. Based on the important role of piRNAs in repressing TEs, we might expect that GSCs and early progenitor cells would have lower levels of TE transcripts than late germ cells. Surprisingly, GSCs and early progeny have higher levels of TE transcripts than late germ cells. One plausible explanation is that GSCs and CBs have a more open chromatin than late germ cells, which results in elevated TE expression and consequently increased piRNA production. Overall, our results establish an interesting baseline for future studies into piRNA dynamics in GSCs and reveal a distinct pattern of piRNA and corresponding TE levels that has not been previously described.

This study has also identified piRNAs derived from 3′ UTRs in both GSCs and SCs. Like transposon-derived piRNAs, 3′ UTR-derived piRNAs can come from sense, antisense, or both strands. Unlike the previous model that 3′ UTR-derived piRNA production is negatively correlated with mRNA expression, our results do not show a clear trend for the relationship between 3′ UTR-derived piRNAs and mRNA levels. In addition, many 3′ UTR-derived piRNAs are differentially expressed in cultured GSCs and SCs. For the 3′ UTR generating both sense and antisense piRNAs, sense and antisense piRNAs do not always correlate in GSCs and SCs. Our results suggest that piRNA production from different 3′ UTRs are regulated differently in GSCs and SCs.

## Alternative promoter usage and splicing likely play an important role in regulating the functions of GSC/progeny and SCs

The germline segregates from the soma during early embryogenesis, and each expresses a unique set of genes that is specifically important for their corresponding lineage development. GSCs and niche cells develop from embryonic germ cells and somatic cells, respectively (Zhu & Xie, 2003; Asaoka & Lin, 2004). As expected, our RNA-seq results have identified GSC-specific and SC-specific genes that fulfill the unique functions of stem cells and niche cells. For example, the piRNA pathway genes implicated in the ping-pong piRNA amplification cycle are only expressed in GSCs, but not in SCs. In addition, we have identified the genes that are expressed in both GSCs and SCs but with different mRNA isoforms. These different isoforms can be generated from the same genes using different promoters in GSCs and SCs. Our RNA-seq results have identified

many genes using different promoters in GSCs and niche cells, such as *tkv* and *nuf*. Some of these mRNAs still have the same coding region but with different 5′ UTRs, which might be important for controlling protein expression levels, whereas other mRNAs encode proteins with different N-terminal sequences, which may provide an additional level of regulation. Furthermore, we have also identified hundreds of genes that exhibit different splicing patterns in GSCs and SCs, yielding mRNAs with different 5′ UTRs, coding regions, and/or 3′ UTRs. The mRNAs with different 5′ UTRs or 3′ UTRs could exhibit changes in mRNA stability, translation, or both. The mRNAs with different coding regions could also give rise to proteins with different functions. Therefore, our RNA-seq results in GSCs and SCs provide a rich resource for other investigators to study the roles of alternative promoter usage and splicing in the regulation of stem cell development.

In summary, this study has used small RNA sequencing to survey the expression of small RNAs, including siRNAs, miRNAs, and piRNAs, in GSCs, early progeny, and SCs. In addition, we have also used mRNA sequencing to carefully examine the expression of TEs and mRNAs in GSCs, early progenitor cells, and SCs. This study has revealed that GSCs and early progeny exhibit more abundant piRNAs and TEs than late germ cells.

# Materials and Methods

### *Drosophila* strains and GSC cultures

The following *Drosophila* stocks were used in this study: $w^{1118}$, $bam^{\Delta86}$ (McKearin & Spradling, 1990), $bgcn^{20093}$ and $bgcn^{20915}$ (Jin et al, 2008), $UASp\text{-}tkv^{M1}$ (Casanueva & Ferguson, 2004), and *nos-gal4* (Van Doren et al, 1998). Flies were maintained and crossed at room temperature on standard cornmeal/molasses/agar media unless otherwise specified. For total RNA isolation, fresh ovaries were dissected in the ice cold Grace Medium.

The *vasa-GFP; hs-bam $bam^{\Delta86}$/$bam^{\Delta86}$* GSC line was established according to the published procedures (Niki et al, 2006). The GSCs cultured in 150-mm culture dishes were dissociated in StemPro Accutase Cell Dissociation Reagent (#A11105-01; Thermo Fisher Scientific) for 5 min and collected by centrifugation at 700g, 4°C for 5 min. The cell pellet was resuspended in PBS, passed through a 22$^1$/2 G needle five times, and filtered with 70-$\mu$m Filcon (340605; BD). GFP-positive GSCs and GFP-negative SCs were purified by sorting dissociated single cells at 20 $\psi$ with 100 $\mu$m tip (InFlux; BD) immediately into TRIzol LS Reagent (10296028; Thermo Fisher Scientific) for total RNA isolation. Before RNA isolation, spike-in RNA standards from ExiSEQ NGS spike-in kit (800100; ExiSEQ) were added to the samples.

### Immunohistochemistry

Immunohistochemistry was performed according to our previously published procedures (Song et al, 2002). The following antibodies were used in this study: mouse monoclonal anti-Hts antibody (1:50, DSHB) and rabbit monoclonal anti-Smad3 (pS423+pS425) antibody (1:100; ab52903; Abcam). All images were taken with a Leica TCS SP5 confocal microscope.

## Small RNA sequencing and mapping

Total RNAs from *Drosophila* ovaries and purified GSCs and SCs were isolated using Trizol, and further purified by organic extraction followed by isopropanol precipitation according to the manufacturer's manual. Small RNA-seq libraries (targeting small RNAs of ~20 nt) were generated from 1 µg high-quality total RNA, as assessed using the Agilent 2100 Bioanalyzer. Libraries were made according to the manufacturer's directions for the TruSeq Small RNA Sample Preparation Kit (RS-200-0012; Illumina). Resulting polyacrylamide gel size–selected libraries were checked for quality and quantity using the Bioanalyzer and Qubit Fluorometer (Life Technologies). Equal molar libraries were pooled, requantified, and sequenced as 50-bp single read on the Illumina HiSeq 2500 instrument using HiSeq Control Software 2.0.10.0, 2.0.12.0, or 2.2.58. After trimming adapters from the reads, remaining reads longer than 15 nt were mapped to the canonical *Drosophila melanogaster* TE sequences downloaded from FlyBase (http://flybase.org). The alignment was performed using Bowtie2 v2.2.0 with default parameters (Langmead & Salzberg, 2012). Unmapped reads were subsequently mapped to the dm6 genome using Bowtie v1.0.0 with the options "-v0 -k1 -m1" (Langmead et al, 2009). Multi-mapped reads were retained exclusively for the cluster discovery step. However, for downstream analysis, multi-mapped genomic reads were discarded.

The counts of aligned reads corresponding to putative piRNAs (reads with lengths between 23 and 30 nt) overlapping unique genomic features (e.g., clusters, UTRs, etc.) were computed in R using the Bioconductor package GenomicRanges (Lawrence et al, 2013). Reads were associated with different biotypes depending on where they mapped in the genome. In the analysis, miRNA reads were identified by overlapping sequencing reads that were 24 nt or shorter to the annotated pre-miRNA locations, whereas siRNA reads were identified by overlapping 22-nt or shorter reads to the annotated endo-siRNA locations, or in the case of transposons any reads that were exactly 21 nt in length (Wen et al, 2014). The remaining 23–30-nt reads that did not align to either ribosomal RNA, small nucleolar RNA, small nuclear RNA, or tRNA features were considered piRNAs.

For the small RNA bar plots of piRNAs, 3′ UTRs, and piRNA clusters, the CPM-mapped read values were normalized to the total number of mapped miRNAs, similar to previous publications (Qi et al, 2011; Hayashi et al, 2016), or for the GSC and SC samples, to the effective library size, unless otherwise stated. For the coverage plots of 3′ UTRs and piRNA clusters, a pseudo-count of one was added to the counts in the coverage plots before log-transformation and loess smoothing.

## mRNA mapping

mRNA-seq libraries were generated from 500 ng of high-quality total RNA, as assessed using the Agilent 2100 Bioanalyzer. Libraries were made according to the manufacturer's directions for the TruSeq Stranded mRNA LT Kit (RS-122-2101; Illumina). Resulting short fragment libraries were checked for quality and quantity using the Bioanalyzer or LabChip GX (Perkin Elmer) and Qubit Fluorometer (Life Technologies). Equal molar libraries were pooled, requantified, and sequenced as 50-bp single read on the Illumina

HiSeq 2500 instrument using HiSeq Control Software 2.0.10.0, 2.0.12.0, or 2.2.58. After sequencing, Illumina Primary Analysis version RTA 1.18.64 or 1.17.21.3 and Secondary Analysis version CASAVA-1.8.2 or bcl2fastq2v2.17 were run to demultiplex reads for all libraries and generate FASTQ files. Reads were also first aligned to the FlyBase canonical TE sequences. Unmapped reads were subsequently mapped to the dm6 genome (Ensembl release 87) using TopHat v2.0.9 (with options "–v 2 –a –best –strata"). Gene-level reads were generated by tabulating the number of reads that uniquely overlapped with the collapsed set of exons for each gene. For the mRNA bar plots of transposons and gene expression, the Reads Per Kilobase of transcript, per Million mapped reads (RPKM) or CPM values were computed using the total number of reads mapped to transposons and exonic regions in the genome.

## DGE analysis

Unless otherwise stated, all differentially expressed features were defined using the Bioconductor package edgeR using the generalized linear model approach. For mRNA analysis, features exhibiting a CPM greater than 10 in fewer than three samples were excluded from the analysis. The CPM cutoff for excluding features in the small RNA analysis was 20, 700, and, 1,000 for piRNA clusters, antisense 3′ UTRs, and sense 3′ UTRs, respectively. For the mutant whole ovary samples, the normalization factors of small RNA-seq reads were calculated based on miRNA counts per library to account for the high potential of global shifts in piRNA abundance between samples. The two most differentially expressed miRNA were excluded from these counts. For the GSC and SC samples, using edgeR, the normalization factors of small RNA-seq reads were calculated based on small RNA spike-ins, present in each library. In these cases, the total library size further used in generating the bar plots was the effective library size (i.e., the total number of mapped reads multiplied by normalization factors). Significantly differentially expressed features were defined as those exhibiting an absolute fold-change of at least two and a FDR of less than or equal to 0.05. The FDR was calculated using the Benjamini–Hochberg procedure (Benjamini, 1995).

## Cluster analysis

Genomic locations of canonical piRNA clusters were downloaded from previous publications (Brennecke et al, 2007; Malone et al, 2009). Genomic coordinates of each were converted to the current genome (dm6) assembly using the UCSC liftOver tool. Cluster fragments were collapsed and those separated by less than 25 kb were merged and only the largest remaining fragment was kept. Only clusters found on the standard chromosomes were used for downstream analysis. Furthermore, in the DGE analysis, only clusters that did not overlap with any known genomic features were used.

To discover novel clusters, we developed a sliding window algorithm that scanned the genome for regions that contained putative piRNAs, reads ranging in size from 23 to 35 nt. Identified regions within 250 nt of one another were merged. The resulting regions were filtered, keeping only those with at least three unique piRNA reads per kilobase. Of the remaining clusters, neighboring

regions within 1,000 nt of one another were merged. Finally, only clusters greater than 2,000 nt in size were retained for analysis.

### Sequence logos

piRNA sequence logos were generated using the R package ggseqlogo. For each experiment, the starting position of reads identified as piRNA were collapsed and only unique start sites were retained. These sites were extended by 10 nt and their underlying DNA sequences were used to generate each logo.

### GO enrichment

GO Term analysis was performed using the Bioconductor GOstats package (Falcon & Gentleman, 2007). A maximum of 20 GO terms were retained for each comparison.

### Alternative splicing

Changes in exon usage were investigated using the Bioconductor DEXSeq package (Anders et al, 2012). Significant events were identified as those exhibiting an adjusted $P$-value of less than or equal to 0.01 and an absolute fold-change of greater than or equal to log2(3) when comparing GSC and SC samples. Sashimi plots were generated using rmats2sashimiplot.py (https://github.com/Xinglab/rmats2sashimiplot).

## Data Availability

All the *Drosophila* stocks are available upon request, and the GEO accession number for all the sequencing raw data is GSE119862.

## Supplementary Information

## Acknowledgements

We would like to thank M Siomi, Developmental Studies Hybridoma Bank, and Bloomington *Drosophila* Stock Center for reagents; the Xie laboratory members for stimulating discussions; Dr. J Brennecke for valuable comments on the manuscript. This work was supported by the Stowers Institute for Medical Research (T Xie).

### Author Contributions

B Story: data curation, software, formal analysis, investigation, visualization, and writing—original draft, review, and editing.
X Ma: investigation.
K Ishihara: investigation.
H Li: methodology.
K Hall: investigation.
A Peak: investigation.
P Anoja: supervision.
J Park: investigation.
J Haug: supervision and methodology.
M Blanchette: data curation, investigation, and methodology.
T Xie: conceptualization, investigation, methodology, and writing—original draft, review, and editing.

### Conflict of Interest Statement

The authors declare that they have no conflict of interest.

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
