## [Reviewer comments · Life Science Alliance]

Expression of piRNA and Transposable Elements in Drosophila Ovarian Germline Stem Cells

Benjamin Story, Xing Ma, Kazue Ishihara, Hua Li, Kathryn Hall, Allison Peak, Perera Anoja, Jungeun Park, Jeff Haug, Marco Blanchette, and Ting Xie

DOI: 10.26508/lsa.201800211

Corresponding author(s): Ting Xie, Stowers Institute for Medical Research

Review timeline:

Submission Date:	2018-10-09
Editorial Decision:	2018-11-08
Revision Received:	2019-05-22
Editorial Decision:	2019-07-01
Revision Received:	2019-10-01
Accepted:	2019-10-04

Scientific Editor: Andrea Leibfried

Transaction Report:

No Peer Review Process File is available with this article, as the authors have chosen not to make the review process public in this case.

1st Editorial Decision

08 November 2018

Re: Life Science Alliance manuscript #LSA-2018-00211

Dr. Ting Xie
Stowers institute for medical research, Kansas City
1000 E. 50th St.
Kansas City, MO 64110

Dear Dr. Xie,

Thank you for submitting your manuscript entitled "Defining the Expression of piRNAs and Transposable Elements in *Drosophila* Ovarian Germline Stem Cells and Niche Cells". The manuscript has been evaluated by expert reviewers, whose reports are appended below.

As you will see, the reviewers are concerned about the validity of the data at this stage. They think that the piRNAs have not been sufficiently validated as being indeed piRNAs, and they think that some of your results indicate that the experimental set-up (cell sorting) doesn't exclude contamination.

We think that these technical issues currently preclude publication in Life Science Alliance, and although your manuscript is intriguing, we think that the points raised by the reviewers are more substantial than can be addressed in a typical revision period. If you wish to expedite publication of the current data, it may be best to pursue publication at another journal.

Given the interest in the topic, we would be open to resubmission to Life Science Alliance of a significantly revised manuscript that fully addresses the reviewers' concerns (except for further reaching functional insight, which would not be needed) and that is subject to further peer-review. Points 1, 2, 3 of reviewer #1 and major point of reviewer #3 would need to be addressed in a good way and we would need strong support from the reviewers for publication. If you would like to resubmit this work to Life Science

Alliance, please contact the journal office to discuss an appeal of this decision or you may submit an appeal directly through our manuscript submission system. Please note that priority and novelty would be reassessed at resubmission.

Regardless of how you choose to proceed, we hope that the comments below will prove constructive as your work progresses. We would be happy to discuss the reviewer comments further once you've had a chance to consider the points raised in this letter.

Thank you for thinking of Life Science Alliance as an appropriate place to publish your work.

Sincerely,

RE: Life Science Alliance Manuscript #LSA-2018-00211R

Dr. Ting Xie
Stowers Institute for Medical Research
1000 E. 50th St.
Kansas City, MO 64110

Dear Dr. Xie,

Thank you for submitting your revised manuscript entitled "Expression of piRNA and Transposable Elements in Drosophila Ovarian Germline Stem Cells". Two of the original reviewers re-assessed your work - original reviewer #1 declined to re-review it.

As already mentioned to you in a pre-decision consultation, it remains still unclear whether the resource value of your work is sufficient for publication here. As you will see below, reviewer #2 (who also assessed your response to original reviewer #1) points out that the description isn't comprehensive enough to allow others to use the data as a resource. Specifically, the reviewer points out that the piRNA profiles of your library and the previously published one should be shown in detail (positions and amounts) (point 4) and that the common and unique piRNA clusters in the different data sets should be shown (point 1).

You provided a preliminary response to the remaining concerns and we have discussed our decision in light of it, also with reviewer #2. We decided that a version that can more easily be used as a resource by others warrants publication here, and we would like to ask you to further revise your manuscript as you already outlined. Please see also reviewer #2's additional comments below to guide you through this final revision. Furthermore, the following editorial issues need to get addressed:

- please add a callout to figure S2G in the text
- please upload all figures (main and suppl) as individual files and without legends, the legends can remain in the main manuscript docx file
- please add a summary blurb in our submission system
- please link your profile in our submission system to your ORCID iD, you should have received an email with instructions on how to do so

A. FINAL FILES:

B. MANUSCRIPT ORGANIZATION AND FORMATTING:

Sincerely,

3rd Editorial Decision

04 October 2019

RE: Life Science Alliance Manuscript #LSA-2018-00211RR

Dr. Ting Xie
Stowers Institute for Medical Research
1000 E. 50th St.
Kansas City, MO 64110

Dear Dr. Xie,

Thank you for submitting your Resource entitled "Expression of piRNA and Transposable Elements in Drosophila Ovarian Germline Stem Cells". We appreciate the introduced changes and that you made the piRNA data more accessible/available to the reader. It is thus a pleasure to let you know that your manuscript is now accepted for publication in Life Science Alliance. Congratulations on this interesting work.

DISTRIBUTION OF MATERIALS:

Again, congratulations on a very nice paper. I hope you found the review process to be constructive and are pleased with how the manuscript was handled editorially. We look forward to future exciting submissions from your lab.

Sincerely,

Andrea Leibfried, PhD
Executive Editor
Life Science Alliance
Meyershofstr. 1
69117 Heidelberg, Germany
t +49 6221 8891 502
e a.leibfried@life-science-alliance.org
www.life-science-alliance.org